

# Quantifying and analysing the angular momentum in volleyball jump serve during the aerial phase: relationship to arm swing speed

Lingjun Liu[1], Zhenxiang Chen[1,2], Defeng Zhao[1], Zhizong Tan[1] and Yaqian Qi[1]

[1] Shanghai Research Institute of Sports Science (Shanghai Anti-Doping Agency), Shanghai, China
[2] School of Athletic Performance, Shanghai University of Sport, Shanghai, China

## ABSTRACT

**Background:** In volleyball, the jump serve is a crucial and commonly used serving technique. Nonetheless, the angular momentum developed during the jump serve remains unexplored. The objectives of the current study were to determine the angular momentum manifesting during the airborne phase of the jump serve and to analyse the correlations between the angular momentum variables and arm swing speed.

**Methods:** Three-dimensional coordinate data were obtained during the jump serves of 17 professional male volleyball players. Correlation and linear regression analyses were used to identify the angular momentum variables linked to the arm swing speed at ball impact (BI).

**Results:** The arm swing speed at BI exhibited significant correlations with the peak angular momentum of the attack arm ($r = 0.551$, $p = 0.024$), non-attack arm ($r = 0.608$, $p = 0.011$), non-attack leg ($r = -0.516$, $p = 0.034$), forearm ($r = 0.527$, $p = 0.032$), and hand ($r = 0.824$, $p < 0.001$). A stepwise regression model ($R^2 = 0.35$, $p = 0.043$) predicted arm swing speed based on the peak angular momentum of the non-attack leg, forearm, and hand.

**Conclusions:** The study results suggest that during the arm-acceleration phase, (1) increasing angular momentum with the non-attack leg helps maintain aerial body balance, thereby enhancing arm swing execution, and (2) controlling the magnitude and timing of the force exerted by the elbow and wrist is crucial for effectively transmitting angular momentum, contributing to an increase in arm swing speed.

# INTRODUCTION

Serving is an essential skill in volleyball, as it plays a significant role in determining the outcome of a match. It is the only volleyball movement unaffected by the tactics and behaviour of the opposing team (*Papadopoulou et al., 2013*; *Lima et al., 2021*; *Moras et al., 2008*). An accurate and powerful serve not only leads to direct scoring opportunities but also exerts psychological pressure on opponents and disrupts their tactical layout

Corresponding author
Zhenxiang Chen,
291852311@qq.com

(*Lima et al., 2021*; *Melnyk & Liakhova, 2022*; *Bari et al., 2023*). The jump serve, a crucial serving technique, has a higher ball speed and higher direct scoring rate compared to other serving styles (*Moras et al., 2008*; *MacKenzie et al., 2012*). Hence, to comprehend and enhance volleyball game performance, the techniques and mechanics of the jump serve must be studied, as they profoundly affect the game outcome.

The jump serve technique comprises five sequential movements: ball toss, run-up, jump take-off, spike, and landing. Biomechanical research on serving or spiking has mainly focused on the characteristics of motion and muscle exertion of the lower limbs in the last step before take-off (*Tilp, Wagner & Muller, 2008*; *Wagner et al., 2009*; *Fuchs et al., 2019a, 2021*; *Tai et al., 2021*) and the motion of the arm swing during the spike (*Coleman, Benham & Northcott, 1993*; *Serrien et al., 2016*; *Giatsis & Tilp, 2022*; *Bari et al., 2023*; *Irawan et al., 2023*). The results of these studies can aid volleyball players and coaches in designing training programmes. However, a need remains for a more comprehensive analysis of angular momentum in the jump serve technique.

The aerial spike of the jump serve encompasses multiple rotational movements of the trunk, arms, and legs. Compared to the traditional analysis of kinematic variables such as joint angle and angular velocity, angular momentum offers a more comprehensive understanding on the mechanics of this intricate movement. Angular momentum is a physical quantity that describes the rotational momentum of an object, and from a biomechanics perspective, it is mainly generated by the moment of force exerted by each joint (*Hinrichs, 1987*; *Bahamonde, 2000*; *Robertson et al., 2014*). In addition, angular momentum is transferred between adjacent body segments and has been used to study some explosive hitting techniques in sports (*Bahamonde, 2000*; *Martin et al., 2013*; *Izumoto et al., 2020*). For instance, *Izumoto et al. (2020)* studied the golf swing technique by quantifying angular momentum. Their findings revealed a significant correlation between the peak trunk angular momentum in the horizontal plane and the peak club head speed and indicated that increasing the angular momentum transferred to the swinging arms through trunk angular momentum is essential for achieving higher club head speeds. *Martin et al. (2013)* investigated the relationship between the segmental angular momentums and ball velocity in a tennis serve. They found that the segmental angular momentums of the trunk, upper arm, forearm, and hand–racket exhibiting a proximal-to-distal sequence were significantly correlated with the ball speed at various key time points. These scholars analysed the mechanisms of angular momentum transfer between adjacent segments or segment groups, offering valuable insights for athletes to optimise and enhance their overall performance.

The aerial spike motion of the jump serve resembles other explosive hitting techniques, such as the tennis serve, and follows the open kinetic chain principle (*Wagner et al., 2014*; *Reeser & Bahr, 2017*). Angular momentum is sequentially increased as it is transferred from proximal to distal segments or from the upper arm to the forearm to the hand, ultimately achieving a high ball speed. Previous research conducted by *Lima et al. (2021)* on the jump serve showed a significant correlation between the arm swing speed and maximum ball speed. Therefore, increasing the arm swing hand speed is essential to impart greater speed to the ball. However, while executing a spike, the entire body is in the air and

experiences no external force except for gravity. Therefore, a mechanism for the transfer of angular momentum to the hand of the attack arm must be determined. Other segments compensate for the angular momentum produced by the swing arm by reducing their angular momentum, as the angular momentum of the entire body is conserved in the air (*Reeser & Bahr, 2017*). According to the principle of angular momentum conservation, to obtain a considerable angular momentum in one body part, the angular momentum of other body parts in the opposite direction must be increased (*Robertson et al., 2014*). However, no studies have been conducted to quantify and analyse the angular momentum during the aerial phase of a jump serve.

This study investigates the relationships between angular momentum variables and swing hand speed during jump serve aerial spiking. We postulate two hypotheses: (1) Drawing from the open kinetic chain principle, we hypothesise that the peak segmental angular momentums of the upper arm, forearm, and hand of the attack arm manifest in a proximal-to-distal order and correlate significantly with arm swing speed at ball impact (BI). (2) We further hypothesise the existence of several crucial correlations that can accurately predict arm swing speed at BI. Clarifying these points can fill the gaps in the existing knowledge system and offer coaches and athletes valuable basic insights into the mechanics of the jump serve for teaching and training.

## MATERIALS AND METHODS

### Participants

Seventeen male professional volleyball players from local volleyball facilities voluntarily participated in this study (15 right-handed and two left-handed players). Their age, height, weight, and volleyball experience were expressed as mean ± *SD* [range], respectively: $20.70 ± 4.88$ years (17–30 years), $1.94 ± 0.06$ m (1.83–2.08 m), $85.17 ± 10.45$ kg (64–104 kg), and $8.29 ± 3.09$ years (5–12 years). Within their respective teams, nine of them serve as outside hitters, two as opposite spikers, and six as opposites. All players had prior experience with the jump serve, having practised it during training sessions and in national or regional competitions. Consequently, they exhibited proficiency in performing the jump serve. The participants had standard Chinese language skills, allowing them to complete the experiment by following the instructions and providing written informed consent. None of the participants had suffered neuro-muscular or skeletal injuries in 3 months before data collection. The study was conducted following the Declaration of Helsinki and approved by the Shanghai Research Institute of Sports Science Ethics Committee (protocol code LLSC20230015; September 21, 2023). Before the experiment, all participants provided written informed consent to confirm that they understood the purpose and potential risks of the study.

### Experimental setup and data collection

The experimental setup is shown in Fig. 1. The experiment was conducted in a regulation-sized volleyball hall (length: 18 m; width: 9 m), and the height of the net was set at 2.43 m. All participants wore close-fitting clothing and their own sports shoes. We utilised a traditional marker-based 3D motion capture system because of its ability to capture intricate

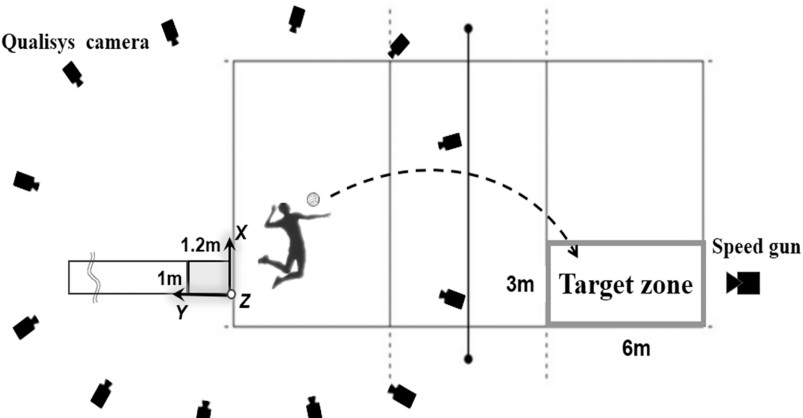

**Figure 1** Experimental setup.

movement details and achieve higher accuracy in spatial coordinate positions (*Scataglini et al., 2024*). A set of 47 markers (diameter: 14 mm) as shown in Fig. 2 were placed on the head vertex, left/right tragus, anterior/posterior shoulders, left/right acromions, suprasternal notch, xiphoid process, 5th cervical vertebrae, 10th thoracic vertebrae, left/right lateral costal borders (10th rib), 3rd metacarpal hands, medial/lateral wrist joints, medial/lateral elbow joints, left/right toes, 1st and 3rd metatarsal, calcaneal tuberosities, medial/lateral malleolus, medial/lateral knee joints, left/right greater trochanters, left/right anterior iliac spines, and left/right superior iliac spines. Three-dimensional coordinate data of the reflective markers during the jump serve were captured at 200 Hz using a 12-camera motion capture system (Qualisys Track Manager, Qualisys, Gothenburg, Sweden). Before each data collection session, the cameras were adjusted and calibrated according to the specific environment of the site. The cameras were arranged as depicted in Fig. 1, with the angle of each camera aligned to ensure coverage of the global coordinate system (L frame). A global coordinate system was set as a right-handed orthogonal reference frame, as shown in Fig. 1. The positive Z-axis was vertically upward, the Y-axis was in the opposite direction of the direction of motion of the spiked ball, and the X-axis was perpendicular to both the Y- and Z-axes.

Before the actual jump serve measurement, the participants were allowed to warm up following their usual routines (*e.g.*, stretching, running, and spikes). Subsequently, the participants were allowed to practice jump serves (approximately 3–5 times), and they were allowed to decide whether they could participate in the formal experiments. A 'T-pose' static trial was captured for each participant, which included placing 47 markers on their body. Each jump serve was executed along a straight line behind the end line of the court. The ready position at which the jump began was decided independently by each individual. At the final step before take-off, both feet had to be planted in an area (length: 1.2 m; width: 1 m) composed of four force plates (FP 600 × 500 × 50, Kistler Group, Winterthur, Switzerland) to confirm that both feet left the ground. The force plates were zeroed repeatedly at the beginning and end of each trial. In addition, a runway made of hard plastic, measuring 6 m in length, 1 m in width, and 0.05 m in height, was installed

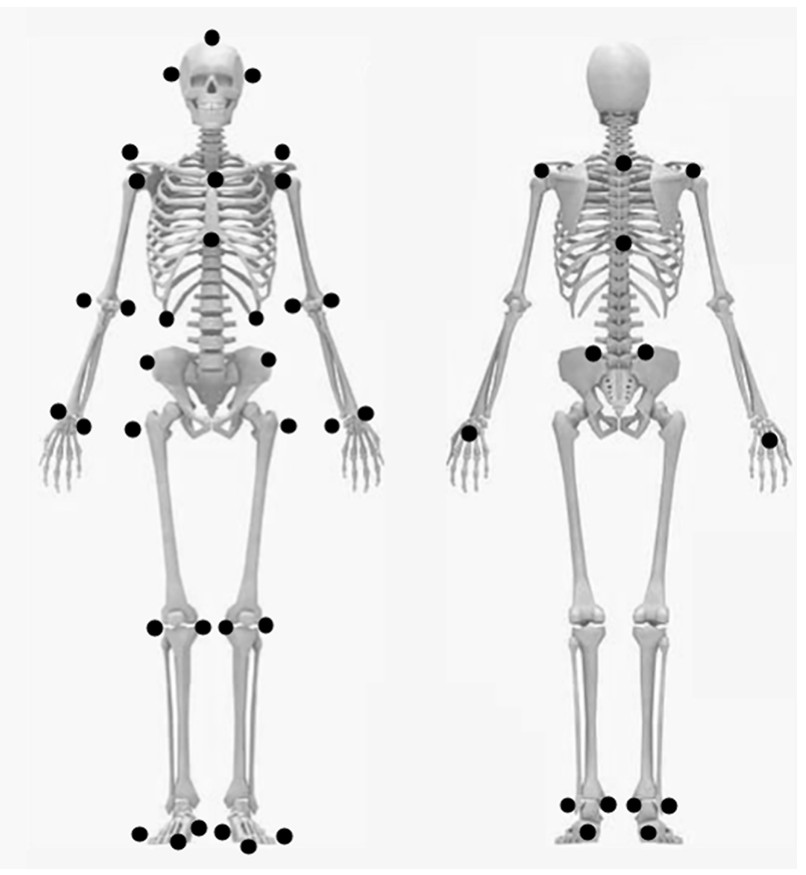

**Figure 2** Locations of reflective markers.

behind the force plates. The participants were required to stand on this runway and then execute a ball toss and run up along the runway.

The participants were instructed to serve a volleyball into the target zone (length: 6 m, width: 3 m) with maximal effort. A successful trial was defined as one in which the ball cleared the net and landed in the target zone. Two experimental assistants were present: the first stood behind the target area, observing whether the ball landed inside or outside the target area, and the other subjectively evaluated whether the ball impacted the net. When a participant completed a successful trial, he was asked to evaluate his athletic performance subjectively using a five-point Likert scale (five points indicating an excellent trial and four points indicating a good trial). The participants were asked to repeat the jump serve until at least three successful trials with scores of four or five points were obtained. The trial with the highest ball speed, treated as the best performance trail, was selected for the subsequent motion analysis. The ball speed was measured using a volleyball pocket radar gun (Smart Coach, Pocket Radar Inc., Santa Rosa, CA, USA), which is the measurement method previously used in volleyball biomechanical studies (*Bujang & Samsudin, 2022*; *Lima et al., 2021*).

## Data processing

All data processing was performed using MATLAB 2016a (MathWorks, Inc., Natick, MA, USA). The raw coordinates of the marker trajectories were smoothed using a low-parm swing speed Butterworth filter, and the cut-off frequency was set to 13 Hz (*Serrien et al., 2016*). The entire body was modelled as a 14-link segment consisting of the trunk, head, upper arms, forearms, hands, thighs, shanks, and feet. The location of the centre of mass (CoM) and moment of inertia of the segment were estimated based on the study by *Ae, Tang & Yokoi (1992)*.

## Airborne phase of jump serve

The airborne phase was defined as the interval between take-off and landing. Take-off (TO) was defined as the instant at which both feet were off the force plates, confirming a vertical ground reaction force (GRF) of less than 5 N. Landing-on (LO) was defined as the time of maximum downward vertical velocity of the CoM. MSER was defined as the timing of the maximum external rotation angle of the shoulder of the attack arm. The instant of ball impact (BI) was adopted from the previous studies (*Wagner et al., 2014*; *Zhang, Sado & Fujii, 2022*), defined as the time at frame 1 (0.005 s) before the instant of detecting the anterior positive horizontal (Y-axis; Fig. 1) acceleration (+) of the CoM of the attack arm hand. A more detailed analysis of the jump serve in the airborne phase was reported by *Reeser et al. (2010)*, who divided the jump serve into the arm-cocking (TO to MSER), arm-acceleration (MSER to BI), and follow-up (BI to LO) phases.

## Calculated parameters

### Arm swing speed

The arm swing speed, as adopted from a previous study (*Tilp, Wagner & Muller, 2008*), was calculated using the square root of the CoM velocity data of the hand of the attack arm, with the value at BI extracted for statistical analysis.

### Angular momentum

The angular momentum was calculated using the methods reported by *Bahamonde (2000)* and *Robertson et al. (2014)*. The whole-body angular momentum was calculated as the sum of 14 body segments. To gain a deeper understanding of the contribution played by each segment in the angular momentum of the whole body, the body was partitioned into segmental systems for a right-handed player: attack arm (right upper arm, right forearm, and right hand), non-attack arm (left upper arm, left forearm, and left hand), trunk-head (trunk, head), attack leg (right thigh, right shank, and right foot), and non-attack leg (left thigh, left shank, and left foot). The units of angular momentum are kg·m$^2$/s. To minimise the effects of differences in height and weight for each participant, all angular momentum variables were adopted from *Hinrichs (1987)*, which were normalised by dividing by the body mass (kg) and square of the height (m) of each participant. Consequently, in this study, the angular momentum variable was expressed in units of s$^{-1}$. Each calculated variable was normalised to 100% of the normalised time. The study focused on the sagittal plane (rotated around the X-axis). The counter-clockwise direction (+) corresponded to

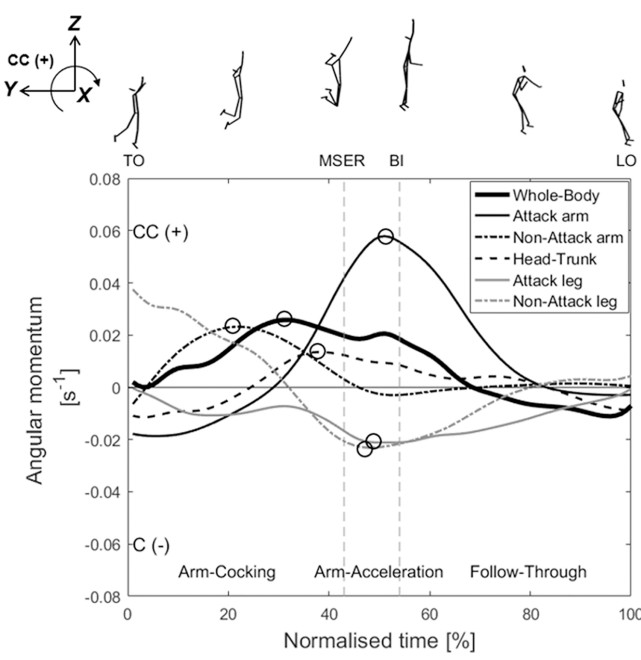

**Figure 3** Mean (*n* = 17) time-series data for the angular momentums of the whole body, attack arm, non-attack arm, head–trunk, attack leg, and non-attack legaround the X-axis during the aerial phase. TO, take-off; MSER, maximum shoulder external rotation; BI, ball impact; LO, landing-on. CC (+), counter-clockwise; C (–), clockwise. The peak angular momentums are marked by black circles (○).

the direction of the spike ball, and the clockwise direction (−) was the opposite of the spike ball direction. For left-handed players, the data were treated as those of right-handed players during analysis and processing.

To investigate the importance of changes in angular momentum on the arm swing velocity during critical temporal phases (arm-cocking and arm acceleration), we employed the analysis method adopted by *Martin et al. (2013)*. The angular momentum variables at TO, MSER, and BI (as shown in Figs. 3 and 4) and the peak angular momentums were extracted for subsequent analysis of their correlation with the arm swing velocity at BI.

### Statistical analysis

The descriptive statistics are presented as mean ± standard deviation (SD). The data normality was checked using the Shapiro–Wilk test. The Pearson product–moment correlation coefficients were used to identify the relationships between the peak angular momentum parameters and the arm swing velocity at BI. Spearman's rank correlation test was conducted when the data were not normally distributed. In interpreting the correlations, the magnitude of *r* was classified as small, 0.1; moderate, 0.3; large, 0.5; very large, 0.7; and extremely large, 0.9 (*Hopkins et al., 2009*). Stepwise multiple linear regression analyses were performed, with arm swing speed as the dependent variable and peak angular momentum (which showed significant correlations only) as the independent variables. Statistical significance for correlation and regression analyses was set at $p < 0.05$.

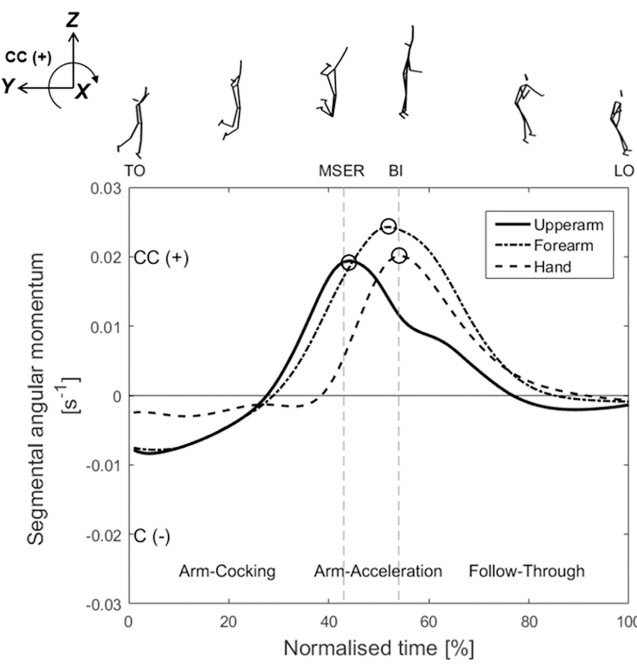

**Figure 4 Mean (*n* = 17) time-series data for the segmental angular momentum of the upper arm, forearm, and hand of the attack arm around the X-axis during the aerial phase.** TO, take-off; MSER, maximum shoulder external rotation; BI, ball impact; LO, landing-on. CC (+), counter-clockwise; C (–), clockwise. Peak angular momentums are marked by black circles (○).

All tests and statistical analyses were performed using the free statistical software JASP (JASP Team, University of Amsterdam, Amsterdam, The Netherlands, 2020).

## RESULTS

The arm swing speeds at BI were 16.08 ± 1.35 m/s, with the value exhibiting a normal distribution. The maximum ball speeds were 27.48 ± 0.85 m/s.

Figure 3 presents the angular momentums of the whole body, attack arm, non-attack arm, head–trunk, attack leg, and non-attack leg. The angular momentum of the whole body gradually increases after TO in the counter-clockwise (+) direction, peaking at 30% normalised time, and then gradually decreases, turning clockwise (–) at 70% normalised time. The angular momentum of the attack arm rapidly increases in the counter-clockwise (+) direction from the later arm-cocking phase and peaks near 50% normalised time. The non-attack arm rapidly increases in the counter-clockwise (+) direction from TO and peaks near 20% normalised time. Then, it exhibits a small clockwise (–) angular momentum during approximately 50–70% normalised time. The head–trunk angular momentum gradually increases in the counter-clockwise (+) direction from the latter part of the arm-cocking phase and peaks near MSER. The statistical results show that the peak angular momentum of the attack arm (0.062 ± 0.017 $s^{-1}$; $r = 0.551$, $p = 0.024$, 95% CI: 0.096 to 0.816, large positive) and non-attack arm (0.024 ± 0.008 $s^{-1}$; $r = 0.608$, $p = 0.011$, 95% CI: [0.18–00.842], large positive) are significantly correlated with the arm swing speed at BI (Table 1). The angular momentum of the attack leg increases after TO in the clockwise (–)

**Table 1 Peak angular momentum variables and correlations with the arm swing speed at ball impact (BI).**

| Angular momentum ($s^{-1}$) | Mean ± SD | r | p | 95% CI | Correlation |
|---|---|---|---|---|---|
| Whole body | 0.030 ± 0.011[#] | −0.240 | 0.352 | [−0.646 to 0.272] | Null |
| Attack arm | **0.062 ± 0.017[#]** | **0.551** | **0.024[*]** | **[0.096–0.816]** | **Large positive** |
| Non-attack arm | **0.024 ± 0.008[#]** | **0.608** | **0.011[*]** | **[0.180–0.842]** | **Large positive** |
| Head–trunk | 0.017 ± 0.007 | 0.162 | 0.535 | [−0.346 to 0.596] | Null |
| Attack leg | −0.026 ± 0.008[#] | −0.051 | 0.846 | [−0.519 to 0.440] | Null |
| Non-attack leg | **−0.026 ± 0.012** | **−0.516** | **0.034[*]** | **[−0.799 to −0.048]** | **Large negative** |
| Upper arm | 0.022 ± 0.006[#] | −0.118 | 0.653 | [−0.566 to 0.385] | Null |
| Forearm | **0.026 ± 0.007[#]** | **0.527** | **0.032[*]** | **[0.062–0.804]** | **Large positive** |
| Hand | **0.022 ± 0.006[#]** | **0.824** | **<0.001[***]** | **[0.568–0.934]** | **Very large positive** |

Notes:
[*] $p < 0.05$.
[**] $p < 0.01$.
[***] $p < 0.001$.
[#] Data not normally distributed.
Bold: indicates the variable is significantly correlated with arm swing speed.

direction, and that of the non-attack leg rapidly increases in the latter part of the arm-cocking phase. The angular momentums of both legs show similar patterns during the arm-acceleration phase, and both peak values appear at approximately 50% of the normalised time (Fig. 3). The peak angular momentum of the non-attack leg (−0.026 ± 0.012 $s^{-1}$; $r = -0.516$, $p = 0.034$, 95% CI: [−0.799 to −0.048], larger negative) is significantly correlated with the hand speed at BI (Table 1).

Figure 4 presents the segmental angular momentums of the upper arm, forearm, and hand of the attack arm. The segmental angular momentums of the upper arm and forearm increase in the counter-clockwise (+) direction from the latter part of the arm-cocking phase, and the peak values of the upper arm appear near MSER and of the forearm at the latter-arm-acceleration phase. The hand angular momentum increases in the counter-clockwise (+) direction from MSER, and the peak value appears near BI. The statistical results show that the peak segmental angular momentums of the forearm (0.026 ± 0.007 $s^{-1}$; $r = 0.527$, $p = 0.032$, 95% CI: [0.062–0.804], large positive) and hand (0.022 ± 0.006 $s^{-1}$; $r = 0.824$, $p < 0.001$, 95% CI: [0.568–0.934], very large positive) are significantly correlated with the arm swing speed at BI (Table 1).

Among the nine independent variables, five correlated with arm swing speed at BI (Table 1). For arm swing speed, the regression model encompassed peak non-attack leg angular momentum, peak forearm angular momentum, and peak hand angular momentum, achieving an adjusted $R^2$ of 0.351 ($p = 0.043$). The equation for arm swing speed is represented as:

Arm swing speed = (14.10 − 3.56 × 10 × peak non-attack leg angular momentum −1.92 × $10^2$ × peak forearm angular momentum + 3 × $10^2$ × peak hand angular momentum).

## DISCUSSION

This study investigated the relationships between angular momentum variables and swing hand speed during jump serve aerial spiking, utilising correlation and regression analyses.

Our results indicate significant correlations between the peak angular momentums of the attack arm, non-attack arm, non-attack leg, forearm, and hand with the arm swing speed at BI (Table 1). Furthermore, the segmental angular momentums of the attack arm, the timing of the appearance of the peak values of the upper arm, forearm, and hand presents a distinct proximal-to-distal sequence (Fig. 4), thereby supporting the first hypothesis. Regression analysis, utilising three independent variables (peak non-attack leg angular momentum, peak forearm angular momentum, and peak hand angular momentum), explains 35% of the variation in predicted arm swing speed. This model successfully predicts 35% of the variation in arm swing speed, thereby supporting the second hypothesis.

During the arm-cocking phase, the decreases in the counter-clockwise (+) (*i.e.*, spike-ball direction) angular momentums of the non-attack arm and non-attack leg compensated for the decreases in the clockwise (−) (*i.e.*, opposite to the spike-ball direction) angular momentums of the attack arm and head–trunk (Fig. 3). Notably, the non-attack arm exhibited a larger counter-clockwise (+) angular momentum, with a significant correlation observed between its peak value (0.024 ± 0.008 s−1) (Table 1). From a volleyball coaching perspective, players need to cock their attack arm and arch their back ('cock the hammer') to prepare to initiate the arm-acceleration phase of the arm swing (*Reeser & Bahr, 2017*). Rotating their attack arm and head–trunk in the clockwise (−) direction as a counter-movement to eccentric pre-lengthening before explosive concentric shortening (*i.e.*, stretch-shortening cycle) can effectively generate considerable momentum to spike the ball. However, these actions can cause the upper body to lean back, compromising balance. It is hypothesised that players initially raise their non-attack arm and then swing it downward (Fig. 3). This downward swing aims to produce significant counter-clockwise (+) angular momentum, counterbalancing the clockwise (−) angular momentum generated by the attack arm. Consequently, the entire body sustains counter-clockwise (+) angular momentum, aligning with the spike-ball direction. This facilitates the execution of the arm swings during the subsequent arm-acceleration phase.

During the arm-acceleration phase, the head–trunk angular momentum increased counter-clockwise (+), peaking near MSER, whereas that of the attack arm rapidly increased in the same direction and peaked near the 50% normalised time (Fig. 3). The peak angular momentum of the attack arm ($0.062 \pm 0.017$ s$^{-1}$) (Table 1) was significantly correlated with the arm swing speed at BI (Table 1). The peak angular momentum of the attack arm appeared considerably later than that of the head–trunk (Fig. 3). Studies on tennis serving (*Martin et al., 2013*) and baseball pitching (*Ramsey & Crotin, 2016*) have suggested that, owing to the shoulder being the terminal point of the trunk, the rotation of the trunk positively impacts the forward rotation of the shoulder, thereby enhancing the speed of the distal segment. Our results support this perspective, indicating that the generated head–trunk angular momentum is effectively transferred to the attack arm, contributing to an increased arm swing speed at BI.

In the later part of the arm-cocking phase, the angular momentums of the attack and non-attack legs simultaneously rotated clockwise (−). The peak angular momentum of the non-attack leg ($-0.026 \pm 0.012$ s$^{-1}$) exhibited negative correlations with the arm swing

speed at BI (Table 1). Our regression analysis indicates that an increase in arm swing speed by 1 m/s corresponds to an increase in clockwise non-attack leg angular momentum by 0.012 s$^{-1}$. The angular momentum generated by the movements of both legs, including hip flexion and knee extension, is consistent with the description provided by *Reeser & Bahr (2017)*. According to the principle of conservation of angular momentum, an increase in angular momentum in one body part leads to increases in the angular momentum of other body parts in the opposite direction (*Robertson et al., 2014*). During the arm-acceleration phase, the angular momentums of both legs increased in the clockwise (–) direction to counteract the considerable counter-clockwise (+) angular momentums of the attack arm and head–trunk. Consequently, the counter-clockwise (+) angular momentum of the whole-body system could be reduced, and the body could be prevented from rotating excessively counter-clockwise (+), keeping the body relatively balanced in the air.

Figure 4 illustrates the segmental angular momentum of the attack arm, encompassing the upper arm, forearm, and hand. The peak counter-clockwise (+) angular momentum of the forearm (0.026 ± 0.007 s$^{-1}$) and hand (0.022 ± 0.006 s$^{-1}$) exhibited significant correlations with the arm swing speed at BI (Table 1). Furthermore, the sequential emergence of peak angular momentum, progressing from the upper arm to the hand, demonstrates a proximal-to-distal sequencing, following the open kinetic chain principle. This sequence effectively enhances the end speed through the transmission of angular momentum from proximal to distal segments in rotational movements, such as tennis serves and baseball pitches (*Sprigings et al., 1994*; *Bahamonde, 2000*; *Martin et al., 2013*; *Ramsey & Crotin, 2016*). Our regression analysis suggests that an increase in arm swing speed by 1 m/s corresponds to a decrease in forearm angular momentum by 0.003 s$^{-1}$ and an increase in hand angular momentum by 0.001 s$^{-1}$. As previously mentioned, angular momentum is primarily generated by the moment of force exerted by each joint. Thus, we recommend that coaches be mindful of controlling both the magnitude and timing of muscle exertion at the elbow and wrist joints of the attack arm. This precise control is essential to optimise the transfer of angular momentum between the segments and to achieve faster arm swing speed.

## LIMITATIONS AND FUTURE WORKS

First, this study did not consider the differences in the arm swing styles of the volleyball players. The end segment speed is mainly produced by the joint and segmental rotation in most explosive-hitting sports. Different types of arm swings may affect the hand speed during BI. Second, in this study, leveraging angular momentum conservation and the open kinetic chain theory, we investigated the correlation between angular momentum and swinging arm speed. Given the primary rotation of body segments around the sagittal plane during the aerial spike motion of the jump serve, our analysis focused specifically on angular momentum within the sagittal plane (rotated around the X-axis); however, the trunk (*i.e.*, pelvic, lumbar, and thoracic regions) during the arm-acceleration phase may exhibit substantial horizontal rotation (rotated around the Z-axis). Future studies could analyse the angular momentum in three dimensions. Third, we observed that the generation of angular momentum from hip flexion and knee extension movements during

the arm-acceleration phase enables the balance of the body in the air to be maintained, which is also essential for the motion of the ball spike. Thus, the kinematic and kinetic characterisation of the three joints of the lower limbs, especially the non-attacking leg, during the aerial phase should be analysed. Finally, this study primarily focused on the jump serve movements of male volleyball athletes and did not include female athletes. Previous studies (*Fuchs et al., 2019a*, *2019b*, *2021*) have highlighted technical and strength differences between the sexes. Therefore, the findings of this study may not be directly applicable to female athletes. This study represents the first biomechanical investigation aimed at quantifying and analysing the angular momentum of the volleyball jump serve. Furthermore, it employs the theoretical principles of angular momentum conservation and transfer to examine the complex coordinated movement of the jump serve. This approach not only bridges a significant gap in the existing knowledge base of volleyball jump serve but also offers volleyball practitioners and coaches deeper insights into the results, thereby enhancing their understanding and application of the jump serve.

## CONCLUSIONS

The main findings can be summarised as follows: During the arm-acceleration phase, both the peak counter-clockwise angular momentum of the attack arm and the peak clockwise angular momentum of the non-attack leg exhibit significant correlations with the arm swing speed at ball impact (BI). Regarding the segmental angular momentum of the attack arm, a distinct proximal-to-distal timing sequence is observed in the upper arm, forearm, and hand. The peak counter-clockwise angular momentum of the forearm and hand significantly correlated with the arm swing speed at BI. A stepwise regression model predicted arm swing speed ($R^2 = 0.35$) by incorporating the peak angular momentum of the non-attack leg, forearm, and hand. These results suggest that during the arm-acceleration phase, players should generate considerable clockwise angular momentum in the non-attack leg to maintain aerial balance and improve the execution of the arm swing. Additionally, precise control over the magnitude and timing of muscle exertion in the elbow and wrist joints of the attack arm can facilitate the effective transmission of angular momentum, thereby enhancing arm swing speed.

## ACKNOWLEDGEMENTS

The authors thank all the volleyball players and researchers for their cooperation.

### Funding

This work was supported by the Shanghai Municipal Science and Technology Commission (No. 22dz1204900), the Shanghai Sports Science and Technology Program (No. 22J013) and the Shanghai Research Institute of Sports Science (Shanghai Anti-Doping Agency) Research Initiation Foundation Program (No. 2023TKS-TYQD001). The funders had no role in study design, data collection and analysis, decision to publish, or preparation of the manuscript.

## Grant Disclosures

The following grant information was disclosed by the authors:
Shanghai Municipal Science and Technology Commission: 22dz1204900.
Shanghai Sports Science and Technology Program: 22J013.
Research Initiation Foundation Program: 2023TKS-TYQD001.

## Competing Interests

The authors declare that they have no competing interests.

## Author Contributions

- Lingjun Liu conceived and designed the experiments, performed the experiments, analyzed the data, prepared figures and/or tables, authored or reviewed drafts of the article, and approved the final draft.
- Zhenxiang Chen conceived and designed the experiments, performed the experiments, prepared figures and/or tables, authored or reviewed drafts of the article, and approved the final draft.
- Defeng Zhao performed the experiments, authored or reviewed drafts of the article, and approved the final draft.
- Zhizong Tan performed the experiments, authored or reviewed drafts of the article, and approved the final draft.
- Yaqian Qi performed the experiments, authored or reviewed drafts of the article, and approved the final draft.

## Human Ethics

The following information was supplied relating to ethical approvals (*i.e.*, approving body and any reference numbers):

Shanghai Research Institute of Sports Science Ethics Committee.

## Data Availability

The analysis data are available in the Supplemental File.

## Supplemental Information

Supplemental information for this article can be found online at http://dx.doi.org/10.7717/peerj.18000#supplemental-information.

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
