# Peer review of "Quantifying and analysing the angular momentum in volleyball jump serve during the aerial phase: relationship to arm swing speed"

_PeerJ, doi:10.7717/peerj.18000_

## Round 0.1 · original submission · Major Revisions

Dear authors,

The study entitled “Quantifying and Analysing the Angular Momentum in Volleyball Jump Serve During the Aerial Phase: Relationship to Arm Swing Speed” demonstrated interesting findings using an appropriate methodological approach. However, some important points must be clarified in the manuscript. Your article has great potential for publication on PeerJ, but the reviewers have requested substantial changes to be made.

Please ensure that all review, editorial, and staff comments are addressed in a response letter and that any edits or clarifications mentioned in the letter are also inserted into the revised manuscript where appropriate.

Reviewer 1 ·

Basic reporting

No comment

Experimental design

No comment

Validity of the findings

The authors should pay attention to this part

Additional comments

Line 40. Maximum shoulder external rotation – MSER
Line 123-161. List the scientific studies that confirm the reliability and validity of diagnostic equipment with which the testing was carried out.
Line 167-169. Centre of mass and Centre of gravity. Marquez et al. (2009) in the study talk about: centre of gravity. Its not good reference for your study. Check this in the full article.
Line 178-181. In this part of text you should to put figure 2 and 3.
Line 184-185. Put study about reliability and validity.
Line 189-198. Delete the specified formulas. It's not necessary.
Line 206. In text: Hinrichs, (1987); In references: Hinrichs, (2016).
Line 214; 218; 223. Arm swing speed. Its wrong. You should to put: Arm swing velocity. Correct this in article.
Line 275. The aim of study must be same in introduction and discussion.
Show the correlation in a separate table.

Annotated reviews are not available for download in order to protect the identity of reviewers who chose to remain anonymous.

Reviewer 2 ·

Basic reporting

- Language is mostly sufficient; at some points, the consensual notions in biomechanics are not used (see below).
- Literature references have errors, e.g., among others, Hinrichs in text (1987) and reference list (2016) etc...
- Structure is good, figures are lacking high spatial resollutions. Tables full of correlation values where a directional regression model should be used are unnecessary.
- Hypotheses should be clearly theory-driven, relevant to the aim of the study and specifically tested for.

Experimental design

- Research question should be theory-driven and directionally stated (e.g., as regression models). Why and to which extent is parameter A impacting upon parameter B?
- Biomechanical motion-capturing is a valid tool, but the parameters used should be in line with common mechanics nomenclature, e.g., angular momentum L is I*w, so the unit is kg*m2/s (kilogram metres squared per second) or Nm. It is not comprehensible why 1/s is used, plus, this study is more about the whole system of angular momenta in the airborne phase of the jump (for a given object or system isolated from external forces, the total angular momentum is a constant). The authors should clearly explain why they use angular momentum (and not another kinematic variable such as angular speed).
- The model tested should be specified and the statistical methods should clearly match these objectives.
- Be much more precise in giving the details about your procedures in every aspect. Also, refer to the motion capturing marker system as used in the literature.

Validity of the findings

- Study is interesting, but the theoretical and statistical approach needs to be improved (see above).
- In this vein, the data lack specifity and clarity. Is feels like cherry-picking and being eclectic.
- Give more information of the players and also more data on inter-individual differences and on individual data.
- Please pronounce your limitations (only sagittal plane etc.) and why this is relevant for your conclusions! Be a bit more cautious wehn interpreting your findings.
- What about female players? At least, can you speculate?

---

## Round 0.2 · accepted · Accept

Dear Author,

Congratulations! After your diligent work addressing the reviewers' comments, I am pleased to inform you that your manuscript has been accepted for publication in PeerJ. This version is more concise and formal, enhancing clarity and flow.

Reviewer 1 ·

Basic reporting

No

Experimental design

No

Validity of the findings

No

Additional comments

Dear Editor,

I wrote a couple of suggestions that the authors should correct before accepting the article.

Annotated reviews are not available for download in order to protect the identity of reviewers who chose to remain anonymous.